# Peer review of "Danazol in Refractory Autoimmune Hemolytic Anemia or Immune Thrombocytopenia: A Case Series Report and Literature Review"

_pharmaceuticals, 2022, doi:10.3390/ph15111377_

Round 1

Reviewer 1 Report

Hi,

I have been concerned about the small number of samples and the brief time point, like 6 months of the study. 

The author put these issues in the limitation of the study, so there is no further comment on this topic.

Could you explain how these drugs work during treatment and any marker that leads to any molecular mechanism? Please add them to the abstract and discussion part.

Thank you

Author Response

Dear reviewer:

Thank you for providing these insights. We have revised our manuscript per your suggestions.

Point 1: Could you explain how these drugs work during treatment and any marker that leads to any molecular mechanism? Please add them to the abstract and discussion part.

Response 1:

 ‘Despite of several hypotheses had been postulated, mechanisms underlying the immune modulation of danazol are under investigation but remain obscure. It appears that danazol has an immune modulator effect by increasing T-helper lymphocytes [31] and influencing Fcγ receptors in monocytes, decreasing platelet destruction [32]. Ahn et al. illustrated that danazol could protect erythrocytes from hypotonic osmotic lysis at low concentrations in an in vitro study and proposed that danazol is incorporated into erythrocyte lipid layers in a reversible manner, expanding their surface area to be resistant to osmotic lysis. Also, danazol increased formed extra folds on erythrocyte membranes and decreased erythrocyte osmotic fragility was noted one month after danazol therapy for ITP patients [33]. Horstman et al. indicated danazol could protect opsonized platelets against complement-mediated lysis [34]. In summary, it seems that danazol not only has an effect in modulating the function of T-helper lymphocyte but in stabilizing the opsonized membrane of erythrocyte or platelet in AIHA or ITP patients against lysis. However, at present, no single detectable marker that leads to the above mechanism is available’  Per your suggestion, we have added the above abbreviated and underlined paragraph to the abstract section and the full underlined paragraph to the discussion section on page 7th at line 18-29th in the revised manuscript, respectively.

31.

Mylvaganam, R.; Ahn, Y.S.; Harrington, W.J.; Kim, C.I. Immune modulation by danazol in autoimmune thrombocytopenia. Clin Immunol Immunopathol 1987, 42, 281-287, doi:10.1016/0090-1229(87)90016-x.

32.

Schreiber, A.D.; Chien, P.; Tomaski, A.; Cines, D.B. Effect of danazol in immune thrombocytopenic

purpura. N Engl J Med 1987, 316, 503-508, doi:10.1056/nejm198702263160903.

33.

Ahn, Y.S.; Fernandez, L.F.; Kim, C.I.; Mylvaganam, R.; Temple, J.D., Jr.; Cayer, M.L.; Harrington, W.J. Danazol therapy renders red cells resistant to osmotic lysis. Faseb j 1989, 3, 157-162, doi:10.1096/fasebj.3.2.2914627.

34.

Horstman, L.L.; Jy, W.; Schultz, D.R.; Mao, W.W.; Ahn, Y.S. Complement-mediated fragmentation and lysis of opsonized platelets: ender differences in sensitivity. J Lab Clin Med 1994, 123, 515-525.

Reviewer 2 Report

Dear authors,

you present interesting results about the use of danazol in AIHA and ITP.  However, in the era of novel treatments I wondering if danazol is the best approach especially for patients with SLE. In your case series, most patients have autoimmune thrombocytopenia, do you have any data concerning bone marrow smears and karyotypes ?

Author Response

Dear reviewer:

Thank you for providing these insights. We have revised our manuscript per your suggestions.

 Point 1: However, in the era of novel treatments I wondering if danazol is the best approach especially for patients with SLE.

 Response 1:

Exactly, as you mentioned ‘in the era of novel treatments I wondering if danazol is the best approach especially for patients with SLE.’ In the current report, we did not mention that danazol therapy is the best approach for SLE-related AIHA or ITP. We only mentioned ‘However, whether SLE with AIHA or ITP is more responsive to danazol therapy needs further investigation’ at the 7th paragraph line 38-42th in the section of discussion and ‘In summary, a relatively lower initial daily and maintenance dose of danazol is a favorable alternative for treating refractory AIHA or ITP with less hepatic AE and has GC sparing effect. The long-term effects of danazol require further prospective and large-sample investigations’ at the last paragraph in the same section of the revised manuscript.

Point 2: In your case series, most patients have autoimmune thrombocytopenia, do you have any data concerning bone marrow smears and karyotypes ?

Response 2:

‘Bone marrow study had been performed for all ITP patients before danazol therapy but only 2 biopsies were performed in this institution with available reports. One pathological report of a patient with SLE and pancytopenia showed severe megakaryocytic hypoplasia, consisting of aplastic anemia, and the other report of a patient with ITP disclosed essentially normal marrow, no increase of plasma cells; consider monoclonal gammopathy of undetermined significance. Both reports were absent of karyotype analysis.’ The above paragraph was underlined and added to the 9th paragraph at line 27-33th in the section on materials and methods in the revised manuscript.